# Immunosuppressive Cell Subsets and Factors in Myeloid Leukemias

**DOI:** 10.3390/cancers13061203

**Published:** 2021-03-10

**Authors:** Julian Swatler, Laura Turos-Korgul, Ewa Kozlowska, Katarzyna Piwocka

**Affiliations:** 1Laboratory of Cytometry, Nencki Institute of Experimental Biology, 02-093 Warsaw, Poland; j.swatler@nencki.edu.pl (J.S.); l.turos@nencki.edu.pl (L.T.-K.); 2Department of Immunology, Institute of Functional Biology and Ecology, University of Warsaw, 02-096 Warsaw, Poland; ekozlowska@biol.uw.edu.pl

**Keywords:** chronic myeloid leukemia, acute myeloid leukemia, immunosuppression, regulatory T cells, extracellular vesicles, inhibitory receptors, myeloid derived suppressor cells

## Abstract

**Simple Summary:**

Effector immune system cells have the ability to kill tumor cells. However, as a cancer (such as leukemia) develops, it inhibits and evades the effector immune response. Such a state of immunosuppression can be driven by several factors – receptors, soluble cytokines, as well as by suppressive immune cells. In this review, we describe factors and cells that constitute immunosuppressive microenvironment of myeloid leukemias. We characterize factors of direct leukemic origin, such as inhibitory receptors, enzymes and extracellular vesicles. Furthermore, we describe suppressive immune cells, such as myeloid derived suppressor cells and regulatory T cells. Finally, we sum up changes in these drivers of immune evasion in myeloid leukemias during therapy.

**Abstract:**

Both chronic myeloid leukemia and acute myeloid leukemia evade the immune response during their development and disease progression. As myeloid leukemia cells modify their bone marrow microenvironment, they lead to dysfunction of cytotoxic cells, such as CD8+ T cells or NK cells, simultaneously promoting development of immunosuppressive regulatory T cells and suppressive myeloid cells. This facilitates disease progression, spreading of leukemic blasts outside the bone marrow niche and therapy resistance. The following review focuses on main immunosuppressive features of myeloid leukemias. Firstly, factors derived directly from leukemic cells – inhibitory receptors, soluble factors and extracellular vesicles, are described. Further, we outline function, properties and origin of main immunosuppressive cells - regulatory T cells, myeloid derived suppressor cells and macrophages. Finally, we analyze interplay between recovery of effector immunity and therapeutic modalities, such as tyrosine kinase inhibitors and chemotherapy.

## 1. Introduction

Myeloid leukemias constitute a group of diseases that arise due to genetic abnormalities in immature myeloid progenitor cells in the bone marrow, predominantly in adults. 

Similarly to other cancers, development of leukemia is a hierarchical event [1,2,3,4] (Figure 1). It involves a core population of tumor initiating cells (TICs)/leukemia initiating cells (LICs), as well as cancer stem cells (CSCs)/leukemia stem cells (LSCs). In leukemia, TICs originate from hematopoietic stem cells and demonstrate self-renewal capacity, which enables initiation and maintenance of the tumor. TICs have been identified in leukemias, including AML [1,5,6], and showed ability to repopulate in immunodeficient mice, confirming their ability to self-renew. These cells are involved in the first step of tumor development (following appearance of oncogenic mutations) and to disease progression, which is concurrent with establishing molecular heterogeneity. One the other hand, LSCs (CSCs) constitute a population of rare cancer cells that express stem features, including ability to self-renew. LSCs often propagate as a response to chemotherapy and are primarily responsible for establishment of treatment resistance and relapse. Importantly, even though cancer stem cells may derive from tumor-initiating cells, these terms are not interchangeable and more likely reflect a specific cellular state.

Chronic myeloid leukemia (CML) develops due to reciprocal translocation between chromosomes 9 and 22, leading to fusion between *BCR* and *ABL1* genes and formation of fusion *BCR-ABL1* gene and, finally, expression of constitutively active BCR-ABL1 kinase [7]. Successful treatment and disease control of CML have been achieved due to clinical introduction (in 2001) of small-molecule drug imatinib mesylate that hampers kinase activity of BCR-ABL1 [8]. However, additional mutations in *ABL1* and other genes (*ASXL1, RUNX1, IKZF1*, others) and dysregulated DNA repair and JAK-STAT signaling can lead to imatinib-resistance and disease progression [9,10]. On the other hand, genomic landscape of acute myeloid leukemia (AML) is far more complex and heterogenous, with several genomic mutations contributing to disease development. Amongst them, genes with most common driver mutations include *FLT3, NPM1, DNMT3, NRAS, TET2* and *IDH2*, often concurrently mutated [11]. Single cell DNA sequencing has further revealed heterogeneity of AML driver mutations and clonal complexity of leukemic cells in single patients [12]. This heterogeneity in genetic background significantly impedes treatment of AML, which currently is mainly treated with chemotherapy. Novel treatment modalities based on targeting of e.g. FLT3 (FMS-like tyrosine kinase), CD33 or BCL-2 (B cell lymphoma 2) have been introduced, but still fail to achieve efficiency comparable to that of imatinib in CML [13]. Significant challenge in therapy of leukemias is eradication of leukemia stem cells (LSCs), as they present self-renewal capacity and strong drug-resistance. Moreover, LSCs may not be easily distinguishable, do not express exclusive markers, which hampers targeting of these cells.

Initial studies suggested that CSCs in leukemias (LSCs) are restricted to the CD34+CD38- phenotype, shared with normal HSCs (hematopoietic stem cells), though this statement has been redefined [1,14]. In CML, development of CML LSCs from CD34+CD38+ cells has also been reported, indicating possibility of differentiation arrest at the level of progenitors [15]. Thus, combination with other specific surface markers such as CD25, CD26, CD33, CD56, CD36, CD123 (the α chain of the IL-3 receptor), IL-1RAP (interleukin-1 receptor accessory protein) has been proposed to specifically identify LSCs population among all CML cells [14,16]. Importantly, for CML cells such signature presents an almost invariable profile. 

On the other hand, as AML is biologically, molecularly and clinically heterogenous, it shows much higher complexity of stem phenotypes. Firstly, similarly as in CML, AML LSCs can be generated from more mature CD34+CD38+ progenitor cells [15]. Importantly, about 20-25% of AML cases are characterized by absence of neoplastic cells with CD34+ stem cell marker [17]. In these patients, with less than 0.5% CD34+ cells, functionally defined CD34- LSCs exist [18,19,20,21]. Such CD34- LSCs, in addition to stem cell features and functions, possess signature of matured precursors. This suggests that LSCs differentiation arrest can take place either at the progenitor (CD34+ population) or at the precursor-like (CD34- population) stage of the differentiation [17]. Crucially for treatment of AML, presence of CD34- LSCs impedes therapeutic monitoring of the disease and assessment if LSCs have been eradicated. 

LSCs have not been universally identified in AML and most likely are heterogeneous and vary between genetic subtypes and individual patients. Their phenotypic identification should include different marker combinations depending on the AML subtype. Therefore, in addition to CD34, which is not uniquely expressed on AML LSCs, the presence of CD25, CD32, CD38, CD44, CD47, CD96, CD123, CLL-1 (C-type lectin-like molecules-1), TIM3 (T-cell immunoglobulin and mucin-domain containing-3) were proposed as specific for AML LSCs [22,23,24,25]. Importantly, the phenotypic signature additionally depends on the genetic/molecular AML background and is associated with clinical outcomes [26]. 

Lack of completely efficient treatment modalities, together with development of drug resistance, in both CML and AML, has shifted interest of scientists and clinicians towards targeting of leukemic microenvironment, including the immune system. Immunotherapies, especially immune checkpoint blockade and CAR-T (chimeric antigen receptor T cells) based therapies have yielded great efficiency in several solid tumors [27], as well as hematological malignancies, such as B-cell malignancies [28] or lymphoma [29]. Chronic and acute myeloid leukemia cells reside in the specific bone marrow (BM) microenvironment, that provides a favorable (protective from therapy) niche, especially for leukemia stem cells (LSCs), thus it constitutes an attractive therapeutic target. Such microenvironment facilitates relapse following chemotherapy, which mostly kills leukemic cells circulating in blood. The main components of BM include endothelial and stromal cells, osteoblasts, as well as immune cells (Figure 2). These components share similarities and differences between different myeloid malignancies and can differentially promote progression of different leukemias [30,31]. 

Several components of the bone marrow niche facilitate immune evasion, which, similarly to solid tumors, has been recognized as one of the hallmarks of myeloid leukemias. These include factors directly of leukemic origin, such as inhibitory receptors expressed on leukemic cell surface and released soluble factors, extracellular vesicles (EVs), as well as immunosuppressive immune cells subsets. These pathogenic immune cells expand and exhibit increased suppressive activity due to leukemia-derived factors and further inhibit effector arms of the anti-tumor immune response. Here, we will evaluate and describe all these components in the context of immune evasion in chronic and acute myeloid leukemia. 

## 2. Dysfunction of Effector Immunity in Chronic and Acute Myeloid Leukemia

Amongst several immune cell types, CD8+ cytotoxic T cells and natural killer (NK) cells are considered subsets crucial for effective anti-tumor immunity, though during leukemia development they become dysfunctional (Table 1). 

In AML, high percentage of total lymphocytes and high percentage of T cells in the bone marrow correlated with better overall survival. Some studies also suggested a higher absolute number of CD8+ T cells in blood of AML patients compared to healthy controls [32]. Immunophenotyping studies have evaluated functional phenotype of these cells, and peripheral blood (PB) CD8+ T cells at diagnosis in AML patients had an antigen-experienced, terminally differentiated phenotype and expressed markers connected with T cell exhaustion, such as programmed cell death protein 1 (PD-1), 2B4, lymphocyte-activation gene 3 (LAG-3), TIGIT (T cell immunoglobulin and ITIM domain), and senescence marker CD57. On the other hand, in vitro stimulated CD8+ T cells from blood of AML patients showed normal proliferation and cytokine secretion, suggesting that these cells may still be capable of effector function [33]. Other functional experiments have demonstrated that T cells from AML patients fail to form fully functional immune synapse and recruit phosphotyrosine signaling molecules to the synapse [34]. CD8+ T cells in CML exhibit a similar dysfunctional phenotype, with decreased expression of TCRζ (T cell receptor ζ) and limited cytotoxic function and secretion of effector interferon γ (IFN-γ) and tumor necrosis factor α (TNF-α) [35,36]. Effector immune response in CML is further inhibited by downregulated expression of trafficking receptor CD62L, on both CD4+ and CD8+ T cells [37]. Also, NK and invariant NK (iNK) cells are downregulated in newly diagnosed CML patients. These cells lack their typical degranulating capability and downregulate effector receptors, such as natural killer group 2D (NKG2D), killer cell immunoglobulin-like receptor (KIR), NKp30, NKp46 in CML [35,38]. AML cells, in a direct contact manner, can induce decreased expression of natural cytotoxicity receptors (NKp30, NKp44, NKp46) on NK cells [39]. 

A large immunogenomic study has revealed that both chronic and acute myeloid leukemia have significantly lower cytolytic infiltrate in their respective niche (BM) when compared to B cell lymphoma or chronic lymphocytic leukemia. The study underlined that low cytolytic especially occurred in AML with *FLT3* and *NPM1* driver mutations. Interestingly, in myelodysplastic syndrome (MDS)-like subtype of AML, which represents less advanced stage of disease, cytolytic activity was higher [40]. Finally, large multiplex immunohistochemistry studies of bone marrow of AML and CML patients have concluded that majority of immune cell subsets described as anti-tumor/activated are downregulated in leukemic BM, compared to healthy counterparts. These included effector subsets of cytotoxic (CD8+) and helper (CD4+) subsets of T cells that express granzyme B, CD27 or OX40, but also NK/NKT cells, M1-polarized macrophages, activated B cells or myeloid dendritic cells (DC1 and DC2) [41,42]. 

Importantly, myeloid leukemias constitute a type of cancer with one of lowest mutational burdens and thus relatively low number of neoantigens, towards which immune response could be targeted [43]. Nevertheless, several leukemia associated antigens (LAAs) have been identified. These contain both neoantigens (such as different junction peptides derived from BCR-ABL1 fusion protein), as well as antigens derived from overexpressed proteins, such as Wilms’ tumor protein (WT1), preferentially expressed antigen in melanoma (PRAME), proteinase 3 (PR3) or neutrophil elastase (ELA2) [44]. This supports existence of anti-leukemic immunity, though in most cases it is evaded by myeloid leukemia cells. LAA-specific CD8+ cytotoxic T cells have been identified in blood of CML patients [35], though they express PD-1 and seem to exhibit an exhausted phenotype [45]. Immunogenicity of AML or CML may however significantly change due to therapy and clonal evolution of leukemic cells [46]. Recent single-cell DNA sequencing study by Morita et al. has revealed mutational history of AML cells from over 100 patients and differential growth of subclones in xenografts. Most importantly, different subclones expressed different amount of surface proteins, including LSCs markers such as CD34, CD33 and CD123. Finally, different subclones have emerged following therapy with FLT3 inhibitors [47].

Restoration or induction of effector immune response may elicit anti-leukemic effects and disease eradication. Allogeneic hematopoietic stem cells transplantation has been used for treatment of both CML and AML [48], as grafted donor cells induced graft versus leukemia (GvL) effect, including destruction of leukemic stem cells. This effect is largely dependent on alloreactive T cells [38]. Part of the graft versus leukemia effect has been appointed to activity of NK cells, as allogenic NK cells have prevented relapse of AML [49]. Furthermore, chemotherapy and treatment with tyrosine kinase inhibitors (TKIs) can alleviate immunosuppression and induce effector immune response in leukemia [50], which will be evaluated in more detail further. 

In this review, we will describe several mechanisms in which chronic and acute myeloid leukemia evade anti-leukemic immunity, by leading to dysfunction of effector immune cells. These factors and immunosuppressive cells could be targeted by immunotherapy, also in combination with already established chemotherapy and tyrosine kinase inhibitors. 

**Table 1 cancers-13-01203-t001:** Changes observed in phenotype and activity of immune effector cells, CD8+ T cells and NK (natural killer) cells, in acute and chronic myeloid leukemia. While the general observation of decreased effector function is relevant to both CML and AML, different effector mechanisms may be affected in each type of leukemia. Increase of a parameter or expression of indicated proteins is marked with a green, upward arrow, whereas decrease is marked with a red, downward arrow. No observed changes are marked with an equal sign.

Cell Type	Malignancy	Trend	Observation (Number/Functionality)	Reference
CD8+ T cells	AML	**↑**	absolute number in PB	[32]
		**↓**	amount in BM	[42]
		**↑**	terminally differentiated/exhausted PD-1, 2B4, LAG-3, TIGIT, CD57 cytokine secretion	[33]
		**↑**		
		**=**		
		**↓**	formation of immune synapse	[34]
		**↓**	granzyme B, CD27	[42]
	CML	**↓**	TCRζ	[36]
		**↓**	cytotoxicity, secretion of IFN-γ, TNF-α	[35]
		**↓**	CD62L	[37]
		**↓**	granzyme B, CD27, OX40, LAG-3	[41]
		**↑**	CTLA-4, TIM-3, PD-1	
NK cells	AML	**↓**	NCRs (NKp30, NKp44, NKp46)	[39]
		**↓**	secretion of IFN-γ	[51]
	CML	**↓**	amount in PB	[35,38,52]
		**↓**	NKG2D, NKG2A, NKG2C	[50]
		**↓**	degranulating capability (CD107a)	[52]
		**↓**	NCRs (NKp30, NKp46), KIR	[35]

PB - peripheral blood; BM - bone marrow; PD-1 - programmed cell death protein 1; LAG-3 - lymphocyte-activation gene 3; TIGIT - T cell immunoglobulin and ITIM domain; TCR - T cell receptor; IFNγ - interferon γ; TNF-α - tumor necrosis factor α; CTLA-4 - cytotoxic T-lymphocyte antigen 4; TIM-3 - T cell immunoglobulin and mucin domain 3; NCRs - natural cytotoxicity receptors; NKG2D/NKG2A/NKG2C - natural killer group 2D/2A/2C; KIR - killer-cell immunoglobulin-like receptor.

## 3. Leukemia-Derived Factors That Promote Immune Evasion

Several factors and molecules that derive directly from leukemic cells have been shown to promote evasion of effector immune response in myeloid leukemias. These include receptors (such as PD-L1, CD47, TIGIT), that mediate immune dysfunction by direct cell-cell and receptor-ligand interactions, as well as soluble factors (such as IDO (indoleamine 2,3-dioxygense) and arginase) and extracellular vesicles (EVs) secreted by leukemic cells (Figure 3). Secreted factors drive immune effector cell dysfunction without direct interaction of cells and possibly in distant tissues, to promote spreading of leukemia.

### 3.1. PD-1/PD-L1

The programmed cell death protein 1/programmed death ligand-1 (PD-1/PD-L1) pathway is considered to be one of the most important negative regulators of the anticancer immune response. PD-1 (CD279) binds to its two ligands – PD-L1 (CD274) and PD-L2 (CD273). PD-1 is expressed on activated T cells and B cells [53] and its binding to PD-L1 or PD-L2 results in inhibition of T cell activation, induction of T cell anergy, and in some cases – apoptosis [54]. Thus, expression of PD-1 on tumor infiltrating T cells is usually considered a marker of dysfunction or exhaustion [55]. PD-L1 is expressed on many types of cells, including tumor cells. Expression of PD-1 on CD8+ cytotoxic T cells in CML patients was higher than in healthy donors. Also, PD-L1 was upregulated on leukemic cells in CML patients, especially during blast crisis phase [45,56]. First studies on the PD-1/PD-L1 axis in AML found that PD-L1 expression on murine AML cell lines has significantly increased when they were grown in vivo. Importantly, mice with PD-1 knock-out developed better anti-tumor response than wild-type counterparts [57]. PD-1/PD-L1 pathway also plays a major role in regulatory T cell (Treg)-induced immunosuppression in AML [58,59]. More recent evidence shows that high expression of PD-L1 on AML cells is associated with worse overall survival in patients with *FLT3-ITD* and *NPM1* mutations [60]. Several studies highlighted induction of PD-L1 expression after treatment with interferons [61,62], while reduced expression of PD-L1 was observed on AML cells in patients after treatment with mitogen activated protein kinase kinase (MEK) inhibitors [61,63]. Additionally, signal transducer and activator of transcription 3 (STAT3) inhibition in AML mouse model resulted in downregulation of PD-L1 and induction of anti-leukemic immune response [64]. A number of studies have found that microRNA miR-34a, acting downstream of mucin 1, cell surface associated (MUC1) or TP53, can be an important regulator of PD-L1 expression in AML [65,66,67]. Increasing evidence suggests that blocking of the PD-1/PD-L1 interaction in myeloid leukemias might effectively restore immune response against disease [59,68,69].

### 3.2. CTLA-4/B7

Cytotoxic T-lymphocyte antigen 4 - CTLA-4 (CD152) is a molecule homologous to a T cell costimulatory protein - CD28. Both are expressed on T cells and share two ligands present on antigen presenting cells: CD80 (B7-1) and CD86 (B7-2). However, while CD28 facilitates T cells’ effector function by delivering an activating costimulatory signal, binding of CTLA-4 to its ligands provides an inhibitory signal to these cells. CTLA-4 is mainly expressed on activated conventional T cells, as well as regulatory T cells. Although CTLA-4 plays a crucial role in prevention of autoimmunity, its activity can often contribute to evasion of anti-tumor immunity in cancer [70]. Upregulation of CTLA-4 was observed on T cell subsets of CML and AML patients [41,71]. Curiously, also leukemic cells from AML and CML patients can express CTLA-4 [72]. Pistillo et al. showed that it was expressed in 25% to 85% of AML and CML samples investigated [73]. Incubation of CTLA-4-expressing AML cells with its soluble recombinant ligands, r-CD80 and r-CD86, induced apoptosis of leukemic cells [74]. It was reported that AML cells can also overexpress CTLA-4 ligands - CD80 and CD86 - to evade immune elimination. However, expression of CD80 on leukemic cells’ surface is rare [75,76,77]. In a murine model of myeloid leukemia, CD80-CTLA-4 interaction resulted in evasion of CD8+ T cell-mediated immune response [78]. Finally, overexpression of both B7 proteins on AML cells is linked to poor prognosis [75,79], and blocking of CTLA-4/B7 interaction seems to be a promising strategy for therapy of myeloid leukemias. 

### 3.3. TIM-3/Galectin-9

TIM-3 (T cell immunoglobulin and mucin domain 3) is another checkpoint molecule involved in immune evasion in leukemia. It can be expressed on CD4+ and CD8+ T cells, Treg (regulatory T cells) or NK cells [80]. In terms of immunosuppression in cancer, galectin-9 (Gal-9) is the most important TIM-3 ligand. In CML, increased expression of TIM-3 was detected on NKT cells [81]. Bruck et al. identified increased expression of TIM-3, PD-1 and CTLA-4 on CD4+ and CD8+ T cells in the bone marrow of CML patients, compared with control samples [41]. Similar trend has been observed in AML [82,83] and was associated with poor survival of patients. Presence of PD-1hi TIM-3+ T cells correlated with leukemia relapse in AML patients post allogeneic stem cell transplantation [82]. Also, AML patients who did not respond to chemotherapy had higher amount of galectin-9 on CD34+ blasts and TIM -3 on T cells than patients in complete remission. This observation indicates that combination of chemotherapy with targeting TIM-3/Gal-9 axis could be an effective therapeutic approach [84]. Interestingly, it was demonstrated that TIM-3+ NK cells produced more IFN-γ after interaction with galectin-9 on AML cells. However, as a consequence, this led to an increased expression of IDO enzyme by AML cells and down-regulated degranulating activity of NK cells [85]. Several studies have pinpointed TIM-3 as a promising candidate for targeting AML leukemia stem cells, as its higher expression levels were observed on LSCs (defined as CD45+CD34+CD38- [22,86] or Lin-CD34+CD38-CD90- [24]) than on normal HSCs [22,23,24,86,87]. Interestingly, TIM-3 together with galectin-9 produce an autocrine loop [88,89] that can stimulate LSCs’ (defined as CD45+CD34+CD38-) self-renewal via activation of NF-κB and β-catenin pathways. Indeed, inhibition of this pathway induced apoptosis of AML cells [88]. Altogether, these studies suggest that TIM-3 may be considered a LSC-specific surface molecule in AML and targeting of the TIM-3/galectin-9 axis could be effective in combination with chemotherapy.

### 3.4. TIGIT/CD155, CD112

T cell immunoglobulin and ITIM domain (TIGIT) is a novel immune checkpoint molecule expressed on activated T cells, Treg and NK cells. It has two ligands – CD155 (PVR, poliovirus receptor) and CD112 (poliovirus receptor-related 2 – PVRL2). Similarly to CTLA-4, these ligands also bind a costimulatory molecule – CD226. Exhausted CD8+ T cells in AML patients exhibit high expression of TIGIT which associates with primary refractory disease in these patients. Simultaneously, CD226 was downregulated on T cells [90]. Interaction between CD226 and its ligands - CD155 and CD112—seems to be crucial for cytotoxic activity of NK cells in myeloid leukemias, as it promotes NK cell adhesion to AML cells and enables leukemic cell killing. To evade this, AML cells express low levels of CD155 and CD112 which results in attenuation of NK cell cytolytic towards AML blasts [91,92]. Similarly, TIGIT expression was observed on CD57+ NK cells in CML which might be a reason of NK cell dysfunction in CML [93]. Similar results were obtained in studies on TIGIT in AML [94,95]. Contrary to previous findings, Stamm et al. reported that TIGIT ligands - CD155 and CD112 are highly expressed on AML cells and blockade of TIGIT-CD155/CD112 axis significantly increased lysis of leukemic cells in vitro in a culture with healthy whole PBMCs (peripheral blood mononuclear cells) [96]. Despite recent advances, still little is known about TIGIT-CD155/CD112 axis in myeloid leukemias and additional studies are needed to implement a novel therapeutic strategy with the use of blocking antibodies.

### 3.5. LAG-3/MHC-II

Lymphocyte-activation gene 3 (LAG-3) is a surface molecule expressed on T cells and NK cells which binds major histocompatibility complex II (MHC-II) with a greater affinity than CD4, and thus negatively stimulates T cell activation [97]. It is suggested that tumor cells express MHC-II to bind with LAG-3 and inhibit TCR signaling on effector T cells. As LAG-3 is overexpressed on T cells in CML and AML patients [71,81], it could be yet another way to evade immune response by leukemic cells [98]. 

### 3.6. CD47/SIRP-α.

CD47 is a transmembrane protein expressed on various types of cells and enables binding of its receptor – signal regulatory protein-alpha (SIRP-α), that is present on macrophages. These interaction leads to inhibition of phagocytosis of CD47+ cells, thereby CD47 has been well recognized as a “don’t eat me” signal. CD47 is upregulated on HSCs during their migratory phase, to protect these cells from phagocytosis [99]. It has become evident that tumor cells also overexpress CD47 to avoid phagocytosis, including in murine and human myeloid leukemias [91,99,100]. Expression of CD47 on human AML LSCs was higher than on normal BM HSCs (Lin-CD34+CD38-CD90+) or multipotent progenitors (MPPs, Lin-CD34+CD38-CD90-CD45RA-) and was associated with the *FLT3-ITD* mutation [101]. Importantly, the level of CD47 expression seems to be patient-dependent [102,103] and extremely high expression of CD47 on AML cells was detected in about 25% of patients’ samples [101,104]. Crucially, an increasing number of studies have found that overexpression of CD47 on AML cells is linked to poor clinical outcomes. Moreover, disruption of CD47/SIRP-α axis impaired AML engraftment and enhanced phagocytosis of leukemic cells both in vitro and in vivo [101,105,106,107]. 

### 3.7. Indoleamine 2,3-Dioxygenase 1 (IDO)

Indoleamine 2,3-dioxygenase 1 (IDO) is an enzyme that catalyzes conversion of tryptophan to kynurenine in the kynurenine pathway and can thus modify immune response by several different mechanisms. Firstly, increased degradation of tryptophan leads to inhibition of the mammalian target of rapamycin (mTOR) pathway and activation of GCN2 (general control nonderepressible 2) kinase in effector T cells, inducing T cell anergy and increased suppressive activity of regulatory T cells (Treg). Moreover, kynurenine binds to aryl hydrocarbon receptor (AHR) to negatively regulate dendritic cell immunogenicity [108]. Vonka et al. assessed IDO activity in CML patients’ samples by measuring the kynurenine/tryptophan ratio in serum, demonstrating a correlation between increased kynurenine levels and leukemia burden which decreased during therapy with interferon γ [109]. In AML, IDO-expressing leukemic cells contributed to an increased number of CD4+ CD25+ Foxp3+ cells among T cells co-cultured with AML blasts. Moreover, these Treg retained suppressive activity, and conversion of CD4+ CD25- T cells to CD4+ CD25+ Treg was abrogated by an IDO inhibitor [110]. A number of studies have found that in both CML and AML increased IDO levels are interferon-induced [109,111,112,113]. It has been suggested that inhibition of a different enzyme – cyclooxygenase 2 (COX2), can restrain IDO-mediated immune dysfunction [112]. The link between increased level of IDO in AML cells and poor prognosis of patients with AML was presented also in other works [114,115,116,117], and this suggests that IDO inhibitors might facilitate therapy of AML through increasing anti-tumor immunity. 

### 3.8. Arginase

L-Arginine is a semi-essential amino acid that plays an important role in cell proliferation and protein synthesis. It is synthesized from L-citrulline and converted to L-ornithine and urea by enzyme arginase. There are two isoforms of this enzyme in mammalian cells – arginase-1 and arginase-2. It is known that cancer cells require higher amounts of nutrients than normal cells and become auxotrophic to some amino acids, including arginine. Though arginine deprivation in tumor microenvironment can make cancer cells more vulnerable [118], depletion of arginine can even more significantly affect T cell activity and leads to polarization of monocytes towards an immunosuppressive phenotype in AML [119]. Increased arginase activity results in impaired T cell proliferation and decreased cytokine production, leading to inhibition of immune response [120]. Several studies have shown that CML and AML patients had higher transcript levels of arginase in peripheral blood, as well as higher level of circulating protein in plasma/serum (compared to healthy counterparts) [56,119,121]. Arginase-1 is considered to be a suppressive molecule released not only by leukemic cells, but also by immune myeloid cells. Concentration of this enzyme in patient plasma correlated with the number of CD11b+ CD14- CD33+ myeloid cells [122]. On the other hand, Mussai et al. demonstrated that AML cells depend on extracellular arginine and import this amino acid using cationic amino acid transporter 1 (CAT-1) and cationic amino acid transporter 2b (CAT-2B) transporters. They also showed that arginine depletion leads to necrosis of AML blasts [123]. Therefore, taking into account an undoubtedly important role of arginine in immune response, researchers should find an appropriate solution for maintaining low levels of arginine in leukemic microenvironment without impairing anti-tumor immunity. 

### 3.9. Leukemic Extracellular Vesicles (EVs) and Non-Coding RNAs.

Extracellular vesicles (EVs) constitute a relatively novel type of intercellular communication – they are small vesicles surrounded by lipid bilayer and can contain proteins, lipids and nucleic acids. As they are released outside cells, they can pass on information either locally or between distant tissues. Several types of EVs, different in size and origin, have been identified, such as exosomes (originating from endocytic recycling pathways and released from multivesicular bodies), microvesicles (released by budding from the plasma membrane), apoptotic bodies or large oncosomes [124]. 

Several studies have implicated EVs in biology of myeloid leukemias and modulation of the bone marrow microenvironment. CML EVs have been shown to promote leukemic growth in an autocrine manner [125], as well as promote and “transfer” resistance to tyrosine kinase inhibitors (TKIs), when released by imatinib-resistant cells [126]. CML-derived EVs contain BCR-ABL1 protein [127] and DNA [128] and can thus lead to development of CML-like disease, as shown after transfer to immunodeficient mice [128]. Similarly, AML-derived EVs have been shown to promote growth of leukemic cells, as well as transport molecules responsible for chemoresistance of AML cells [129,130]. Both AML-and CML-derived EVs modulate components of the bone marrow microenvironment – they inhibit osteogenesis and normal hematopoiesis [131], modulate bone marrow stromal cells [132,133,134] and facilitate angiogenesis [135], all to promote leukemia development. 

EVs have been already well recognized as active participants in the immune response [136] and immune evasion in different cancers [137], including myeloid leukemias. AML-derived EVs can carry immunosuppressive protein cargo, such as TGF-β (transforming growth factor β), PD-L1/PD-1, Fas/FasL, ectonucleosidases CD39 and CD73 [138] and “don’t eat me” signal protein CD47 [139]. This immunosuppressive cargo, especially TGF-β, led to dysfunction of NK cells, by downregulating their cytotoxicity, NKG2D expression and migration towards AML cells [138,140]. Leukemic EVs can also regulate T cell responses – EVs released in vivo in mice by AML- patient-derived xenograft (PDX) cells and CML cell lines induced apoptosis of CD8+ effector T cells [141,142]. Our studies showed that CML-derived EVs controlled biology of Treg, as they increased suppressive function and forkhead box P3 (Foxp3) level in Treg, which are beneficial for leukemia progression [143]. Couple of works demonstrated strong interaction between leukemic extracellular vesicles and different subsets of myeloid cells. In both CML and AML extracellular vesicles were shown to promote differentiation and expansion of functional myeloid derived suppressor cells (MDSC) [144,145,146]. The mechanism is yet to be fully described, though several signaling pathways and factors were shown be involved, such as MUC1 in AML cells [146] and activation of Akt/mTOR pathway via TLR2 (toll-like receptor 2) in MDSC [145]. EVs derived from K562 CML cells influenced phenotype of macrophages, by inducing secretion of interleukin 10 (IL-10) and TNF-α and decreasing release of NO (nitric oxide) and expression of inducible nitric oxide synthase (iNOS), suggesting possible polarization towards an immunosuppressive phenotype [147]. EVs from serum of AML patients differentially affected dendritic cells – though they decreased effector function of mature dendritic cells, they increased functionality of immature dendritic cells. They could thus induce either suppressive or stimulating effect in terms of anti-leukemic immunity mediated by dendritic cells [148].

Other crucial cargo contained and shuttled via EVs is different types of non-coding RNAs, such as microRNAs (miRNAs), lncRNAs (long non-coding RNAs) or Y-RNAs [149]. These can be shaped by immune stimuli in the microenvironment and thus constitute immune mediators [150]. Little is known about these molecules in myeloid leukemias. Recently, miRNAs have been implicated in pathology of leukemias, as e.g. miR-300 was shown to maintain quiescence and drug-resistant phenotype of LSCs in CML [151]. On the other hand, BCR-ABL-repressed miR-139-5p has been shown to downregulate leukemic cells’ proliferation, with an anti-leukemic effect [152]. miRNAs, such as CML LSC-supporting miR-126, are often shuttled via EVs (miR-126 from endothelial cells) [153], increasing their relevance and applicability for biomarker studies. High throughput studies have identified miRNAs and lncRNAs enriched in AML cells, and revealed that these may be involved in modulating both T cell [154] and NK cell responses [155]. Interestingly, miR-300 was crucial not only for LSCs pro-leukemic properties, but also impaired NK cells effector function in CML. This miRNA was transported via EVs derived from mesenchymal/stromal cells, revealing complexity of the microenvironment and wide influence of single miRNAs [151]. PD-L1 expression on AML cells in under negative regulation by miR-34a [65] and low expression of miR-34a has correlated with high PD-L1 levels in AML patients [67]. On the other hand, miR-34a expression was repressed by MUC1 in AML cells. MUC1 targeting and increase in miR-34a levels led to decrease of PD-L1, presenting an attractive therapeutic target in AML [66]. Finally, miR-21 in AML-derived EVs was shown to induce apoptosis of effector T cells, but also increased expression of immunosuppressive genes (*Il-10, Foxp3, Il-4*) in T cells [142].

Extracellular vesicles thus constitute another important factor significant for development of myeloid leukemias. Importantly, due to easy access to EVs from blood and increasingly improving methods of their analysis [156], these particles and their content could potentially be used as a liquid biopsy in leukemias, as indicators of disease relapse or development. 

## 4. Regulatory T Cells in Myeloid Leukemias

Regulatory T cells (Treg) are a subpopulation of CD4+ T cells, characterized by high levels of CD25 (in humans additionally low CD127) and expression of transcription factor forkhead box P3 (Foxp3). They can differentiate either in the thymus (thymic Treg), to establish immune tolerance towards self-antigens, or in the periphery (peripheral Treg), to e.g. establish tolerance towards commensal microbiota or facilitate cancer progression. In the latter case, they thus play a pathogenic function, whereas in other cases mentioned above Treg are beneficial and help maintain homeostasis [157,158]. In tumors, Treg polarize towards a highly suppressive, effector phenotype, to promote progression of cancer and metastasis. In the past few years, several genomic studies have identified several receptors that determine this phenotype, such as CCR8 (chemokine (C-C motif) receptor 8), inducible T-cell costimulator (ICOS), 4-1BB, TIGIT, CTLA-4 and others [159,160,161], as well as molecular regulators of tumor effector phenotype of Treg, such as recently identified interferon regulatory factor 4 (IRF4) [162]. Multiple studies have demonstrated that Treg, as well as Foxp3 itself, are under complex molecular control, via transcription factors and conserved non-coding regions in the Foxp3 promoter, as well as epigenetic regulators that imprint specific epigenetic signature of Treg subsets [163,164]. All recent discoveries in the Treg field demonstrate very complex biology of these cells, both at molecular and cellular level. Contrary to “traditional” view of Treg as cells driven solely by Foxp3, Treg can polarize into different effector phenotypes, quite often tissue dependent [165]. Here, changes in phenotype and function of these cells will be discussed in the context of developing myeloid leukemias and their bone marrow niche (Figure 4). 

In the past 10 years it has been well established that amount/percentage of regulatory T cells is increased in both chronic and acute myeloid leukemia. Percentage of Treg (identified as CD25hi CD127lo cells) among T cells is increased in peripheral blood (PB) of AML patients [166,167]. CD25hi Treg from PB had increased suppressive activity and expression of molecules such as glucocorticoid-induced TNFR-related protein (GITR), CTLA-4 and granzyme B, but not CCR4 (C-C chemokine receptor type 4), CD39 and CD73 [167]. However, the suppressive activity of AML Treg was observed primarily towards CD4+, but not CD8+, responder T cells. Moreover, not all effector functions of effector (responder) T cells were influenced by AML Treg, as e.g. TNF-α production was not affected, whereas IFN-γ production was downregulated, compared to activity of healthy donor Treg [168]. In blood of AML patients, circulating T follicular regulatory cells (characterized as CXCR5+ PD-1+ Foxp3+) were increased, suggesting increased inhibition of B cell responses [169]. AML Treg are also potent producers of interleukin 35 (IL-35), which was more abundant in plasma of AML patients and could protect AML blasts from chemotherapy-induced apoptosis [170]. Percentage of Treg is also increased in blood of CML patients. Especially effector memory (CD45RO+ CD27-) and terminally differentiated effector memory (CD45RO- CD27-) Treg are expanded in CML patients at diagnosis [50]. In CML, it was demonstrated that CD25hi Treg suppress effector responses (proliferation, IFN-γ and granzyme B production) of T cells specific for LAAs, such as HLA-A3 restricted BCR-ABL peptide [171]. 

Importantly, Treg are also increased in the bone marrow (BM) of AML and CML patients, as revealed by both flow cytometry [168,172] and multiplex histology [41,42]. This included decreased ratio of Th17/Treg cells, confirming polarization of BM microenvironment towards immunosuppression [172]. Treg in the AML bone marrow expressed more of checkpoint molecule OX-40, but not PD-l, TIM-3, LAG3, ICOS or 4-1BB [173]. This increased presence of Treg in the BM is largely due to higher expression of CXCR4 (C-X-C chemokine receptor type 4) and chemotaxis of Treg towards bone marrow cells and BM-derived factors [168]. Indeed, in mouse MLL-AF9 model of AML, blocking of CCL3-CCR1/CCR5 and CXCL12-CXCR4 axes has reduced migration of Treg to the BM and has slowed down progression of leukemia [174]. 

Reports suggest that percentage of Treg at diagnosis of AML can be a prognostic factor of response to chemotherapy, as patients who achieved complete remission had lower percentage of Treg in blood than those who failed to respond [32,167,175]. However, in pediatric AML such prognostic value has not been observed [176]. In CML, amount of peripheral blood CD25hi Foxp3+ Treg correlated with amount of *BCR-ABL1* transcript level, as well as counts of leukemic blasts. Importantly, numbers of Treg were significantly higher in patients in accelerated and blast phase (compared to chronic phase patients), as well as in patients with higher Sokal score [177]. Also increased proportion of Treg in the bone marrow predicts inferior survival of AML patients [42]. Several studies have demonstrated that percentage of Foxp3+ Treg is higher (compared to healthy controls and patients at diagnosis) in AML patients who were refractory to therapy and relapsed [172,173]. Increased amount of Treg (CD4+ CD25hi Foxp3+) was also identified in patients with myelodysplastic syndrome (MDS), correlating with high-risk MDS and higher number of blasts in the BM. This supports significance of Treg in progression of MDS, also towards AML [178]. 

Finally, significance of Treg for progression of myeloid leukemias has been underlined by studies using in vivo mouse models and tracking Treg changes during therapy. In mouse model of MLL-AF9 AML, developed in Foxp3^DTR^ mice (which enable depletion of Treg by diphtheria toxin injection; DTR - diphtheria toxin receptor), depletion of Treg has reduced leukemic burden and prolonged survival of mice [174]. In mouse AML model induced by C1498 cells, Treg have inhibited antitumor function of in vitro expanded, AML-reactive cytotoxic T cells (CTLs), that were transferred into leukemia-bearing animals. Depletion of Treg by IL-2DT (diphtheria toxin conjugated to IL-2) has enabled for transferred CTLs to expand at leukemic sites and stop disease progression [179]. In the same mouse model, depletion of Treg with anti-CD25 antibody has increased efficiency of a dendritic cell-based vaccine against AML [180]. Such experiments have provided a direct proof for immunosuppressive and leukemia-promoting role of Treg. Similarly, clinical reports suggest that initial high numbers of Tregs prevent successful therapeutic outcome of allogeneic stem cell transplantation (SCT) in CML [181] and that depletion of CD4+ CD25+ cells from donor infusions may potentially improve efficacy of SCT without induction of graft-versus-host disease [182]. Finally, in both AML and CML, patients who have responded to chemotherapy/TKIs usually have significantly lower Treg numbers, which will be described in detail in last part of this Article. 

Relatively little is known about mechanisms that drive differentiation and suppressive function of Treg in myeloid leukemias. One of the factors of interest is enzyme IDO (indoleamine 2,3-dioxygenase), which is engaged in tryptophan metabolism and can be engaged in induction of Treg from CD25- naïve T cells. Expression of *IDO* and *FOXP3* has been positively correlated in blood of AML patients [117]. AML cells have been identified as IDO-expressing and in IDO+ AML amount of CD4+ CD25+ cells and *FOXP3* mRNA level were higher. In cocultures of IDO+ AML with CD3+ T cells, CD25+ Treg with suppressive activity have expanded, which was abrogated by IDO inhibitor 1-MT [110]. Also bone marrow mesenchymal stem cells from AML patients are capable of producing IDO, which correlated with amount of CD4+ CD25hi Treg in BM of patients [183]. Several studies have also implicated costimulatory and coinhibitory pathways in control of Treg biology in myeloid leukemias. ICOSL (inducible T-cell costimulator ligand) expressed on AML cells has induced expansion of both human Treg in vitro and in vivo in mouse model of AML, whereas blockade of ICOSL in vivo partially reversed Treg expansion and slowed down AML progression [184]. Similarly, PD-L1 expressed on AML cells can control Treg biology and contribute to generation of PD-1+ positive Treg, that expressed more IL-10 and had higher suppressive activity. PD-L1/PD-1 interaction between AML cells and Treg thereby promoted leukemia development in vivo [59], as well as inhibited effector function of adoptively transferrer cytotoxic T cells [58]. Foxp3+ Treg were also downregulated in Gal-9 knock-out mice engrafted with C1498 AML cells, hinting that galectin-9 (ligand for TIM-3) may be engaged in expansion of Treg in AML [185]. Clinical studies have found correlation between expression of CD200 on AML cells and amount of CD25hi Foxp3+ Treg in blood [186]. Analysis of clinical samples has also revealed a potential role of elevated TNF-α levels in AML and generation of TNFR2(tumor necrosis factor receptor 2)-expressing Treg [187]. Finally, also regulatory B cells (CD19+ CD25hi CD38hi) from AML bone marrow and extracellular ATP (adenosine triphosphate) from chemotherapy-treated, dying AML cells have been implicated in Treg induction [168,188]. Much less is known about mechanisms that drive Treg expansion and function in CML. Our study has pin-pointed CML-derived extracellular vesicles (EVs) as drivers of Foxp3 expression and suppressive activity of Treg in CML [143]. 

## 5. MDSC and Macrophages in Myeloid Leukemias

### 5.1. Myeloid Derived Suppressor Cells (MDSC) in Myeloid Leukemias

Immune myeloid cells are one of the major immune cell subsets demonstrating immunosuppressive potential. They constitute a heterogenous group of cells that include several types of monocytes/macrophages, dendritic cells and granulocytes, that can polarize and differentiate into several subsets, especially in inflammatory conditions or cancer. In cancer, myeloid derived suppressor cells (MDSC) have been well studied as cells that are able to induce immune escape and progression of malignant cells [189]. MDSC are a heterogeneous group of immature myeloid cells, consisting predominantly of granulocytic (G-MDSC) and monocytic (M-MDSC) MDSC and immature myeloid cells (IMCs). Various combinations of surface markers have been used to identify subpopulations of MDSC - M-MDSCs are usually identified as CD14+ CD11b+ CD33+ HLA-DR-/lo cells, whereas G-MDSCs as CD66b+ CD15+ HLA-DR- cells with intermediate expression of CD33 and a variable expression of CD11b, depending on maturation stage. IMCs (immature myeloid cells) are defined as CD11b+ CD33+ CD14- HLA-DR- CD34+. MDSC exert an immunosuppressive effect, especially on effector T cells, through different mechanisms - release of arginase-1 (Arg1), nitric oxidase synthase 2 (NOS2), reactive oxygen species (ROS), cyclooxygenase 2 (COX2), transforming growth factor β (TGF-β) and other immunosuppressive cytokines. They can also mediate immunosuppression indirectly - by upregulation of Treg [190]. 

In peripheral blood of newly diagnosed CML patients, accumulation of MDSC has been described. They derive partially from the tumor clone, showing BCR-ABL1 expression, and partially from healthy myeloid cells that polarized towards an immunosuppressive phenotype (Figure 5) [56]. Importantly, amount of MDSC correlated with disease progression and level of *BCR-ABL1* transcript [190]. Clinical studies showed an accumulation of MDSC in the peripheral blood and bone marrow of AML patients at diagnosis, compared with healthy donors [145,146,191]. MDSC have also been implicated in AML development from MDS and myeloproliferative neoplasms (MPN), as they were reported to suppress the naturally occurring, effective anti-tumor immune response in MPN and MDS patients [192]. An in vitro study confirmed a direct interaction between CML cells and MDSC – M-MDSC stimulated proliferation of CML cells in a co-culture with both K562 cells and primary CD34+ cells from the leukemic bone marrow. In vivo, M-MDSC promoted growth of subcutaneously injected K562 cells, when both cells types were mixed and injected in a Matrigel matrix into mice [193]. Further studies revealed that accumulation of G-MDSC was more consistently pronounced than M-MDSC in CML. Moreover, arginase-1 (Arg1) expression in G-MDSC was higher in CML subjects that healthy subjects. G-MDSC isolated from the blood of CML patients markedly suppressed normal donor T cell proliferation in vitro [121]. Functionally, also MDSC from bone marrow of AML patients exhibited high expression of Arg1 and IDO and efficiently suppressed CD8+ T cells’ effector function [194]. A recent study suggests that CD8+ T cell directed suppressive activity of MDSC in AML is based on VISTA (V-domain Ig suppressor of T cell activation) expression, as this suppressive effect was abrogated by silencing of VISTA in MDSC of AML patients [195].

Alongside creating leukemia-supporting milieu, MDSC may challenge successful therapy. Several studies demonstrated that MDSC negatively affected survival of AML, reported both in a mouse models and in patients. However, in allogeneic hematopoietic cell transplantation these cells were found to be beneficial. They reduced graft vs. host effect, by modulation of alloreactive T cell T function, but, crucially, without interfering with the therapeutic graft vs. leukemia effect [196]. The most recent preclinical study shows an inhibitory effect of MDSC on AML targeting CAR-T-based therapy [197].

Several MDSC inducing mechanisms have been identified in myeloid leukemias, though these are less understood in AML. In CML, different types of extracellular vesicles of leukemic origin were shown to induce MDSC. Small EVs/exosomes from serum of CML patients promoted conversion of peripheral blood monocytes into M-MDSC, but not G-MDSC in vitro. Obtained M-MDSC were fully functional and capable of inhibiting proliferation of autologous T cell [144]. On the other hand, K562-derived microvesicles promoted both G-MDSC and M-MDSC differentiation in a whole PBMC culture and in vivo. Interestingly, this effect was not observed when K562 cells were treated with TKIs (imatinib and dasatinib), but was again pronounced after discontinuation of drug treatment [193]. Also, mesenchymal stem cells (MSC) isolated from the BM of CML patients, but not BM of healthy donors, induced G-MDSC in a coculture with PBMCs. These G-MSDC exhibited superior suppressive activity toward autologous T cells and expressed higher transcript levels of *Arg1, TNF-α, IL 1β, COX2* and *IL-6* than counterparts from co-cultures with MSC of healthy origin. The observed phenomenon could be induced by TGF-β, IL-6 (interleukin 6) and IL-10, as higher transcript levels of these factors was observed in MSC from leukemic donors [198]. Also, AML-derived extracellular vesicles may play a role in expansion of MDSC, in a mechanism dependent of MUC1 (mucin 1, cell surface associated) activity in AML cells [196]. Role of MUC1 activity in regulation of MDSC was further underlined by correlation of amount of MDSC and MUC1 expression in AML patients [191]. Furthermore, the ability of AML-derived EVs to induce MDSC from peripheral blood monocytes was shown to be dependent on the presence of palmitoylated proteins on EVs’ surface. Palmitoylated proteins acted by activation of TLR2 to initiate Akt/mTOR–dependent induction of M-MDSC [145]. 

### 5.2. Macrophages in Myeloid Leukemias

Tissue resident macrophages contribute to tissue formation, metabolism, homeostasis and repair. They are derived from either primitive embryonic macrophages (originating from the yolk sac or fetal liver) or blood monocytes. Macrophages are considered cells with high plasticity. In response to variety of signals from the microenvironment they can polarize into functionally different subsets, both inflammatory (often referred to/generalized as M1) and regulatory/alternatively activated (often referred to/generalized as M2), though this plasticity seems to be mainly attributed to monocyte-derived macrophages [199]. It has been well documented that macrophages can be polarized by solid tumors to a phenotype that facilitates cancer development, progression, and metastasis [200]. However, little is known about the role of macrophages in the development and progression of myeloid leukemias (Figure 5). 

In the BM niche, quiescence of both normal HSCs and malignant LSCs is critically determined by interactions with the cells of BM niche, including macrophages. Both CD169+ and Ly6G+ macrophages have been identified as critical BM niche supportive cells, as experimental depletion of macrophages has shown that they negatively regulate HSC proliferation and pool size. Recently, new pieces of data show that in the spleen, a secondary niche for myeloid leukemia stem cells, LSCs (defined as Lin-c-Kit+Sca-1+) were exclusively localized within the red pulp and in close proximity to macrophages, suggesting the role of macrophages in maintaining the quiescent state of LSCs [201,202,203]. In the bone marrow of untreated CML patients, a higher number of cMAF+pSTAT1−CD68+ M2-polarized macrophages (compared to healthy controls) has been detected [41]. Further analyses of CML bone marrow samples revealed that the percentage of CD68+, CD163+, and CD206+ macrophages in BM samples of CML patients was significantly higher than in the control group and was gradually elevated during the progression of CML from chronic phase to blast crisis. As CD163 and CD206 molecules are markers of alternatively activated/suppressive macrophages, the percentage of such macrophages may be considered a key factor for disease progression and a potential therapeutic target [204]. 

Macrophages isolated from different leukemic mouse models supported in vitro expansion of AML cell lines better than macrophages from non-leukemic mice, in a Gfi1 (Growth factor independence 1)-dependent manner. Also, the grade of macrophage infiltration into the BM correlated with survival of mice with developing acute myeloid leukemia. In AML patients, the percentage of CD163+CD206+ M2-like macrophages in the bone marrow was significantly elevated, compared to healthy controls [205]. Another study confirmed the role of M2-polarized macrophages in AML progression. It showed that low level of the monocytic leukemia zing-finger (MOZ), involved in polarization of macrophages toward an inflammatory phenotype, is associated with poor prognosis [206]. CD206+ macrophages were also indicated as a novel prognostic factor for AML. Patients with AML exhibited increased frequencies of CD206+ M2-like macrophages in bone marrow and level of infiltration of M2-like macrophages positively correlated with poor outcome [207]. 

## 6. Immune Response Recovery after Therapy in Myeloid Leukemias

Multiple studies have documented that, especially in chronic (but also acute) myeloid leukemia, successful therapy using established treatments (such as tyrosine kinase inhibitors - TKIs) occurs concurrently with recovery of effector immune response. This proves the importance of functional immunity in recovery from leukemia.

In CML, treatment with different TKIs led to an increased amount of NK cells in the bone marrow, whereas dasatinib was the only TKI with potent effect on CD8+ cytotoxic T cells [208]. Another study confirmed that dasatinib may have the strongest effect in terms of potentiating the immune response [209]. A more detailed study by Hughes et al. revealed robust restoration of effector immune response in PB of patients that responded to TKIs and achieved major or complete molecular response. Not only amount of effector cells was restored, but also their functionality, such as degranulating capability (CD107a expression) of NK cells, CD8+ T cells response to LAAs in an ELISPOT (enzyme-linked immunospot assay) assay and lower expression of PD-1 on both CD4+ and CD8+ T cells [50]. Also CTLA-4 expression on T cells has decreased after imatinib treatment [210,211]. NK cells from patients who successfully responded to imatinib treatment also had a more mature phenotype and secreted more TNF-α and IFN-γ [212]. In AML patients who responded to chemotherapy, CD8+ T cells have regained a functional phenotype (expression of *ICOS, CD28*, downregulation of apoptotic and inhibitory transcripts), as revealed by transcriptomic analysis [213]. 

Alongside recovered functionality of effector T cells and NK cells, successful therapy leads to decreased numbers and functionality of suppressive immune cells. 

Different subsets of MDSC were affected by TKIs – amount of G-MDSC in blood of CML patients decreased after imatinib, nilotinib and dasatinib and M-MDSC decreased after dasatinib treatment. Same study identified a significant correlation between lower numbers of M-MDSC and major molecular response [144]. Even lower numbers of M-MDSC were observed in patients with complete molecular response (CMR) [211]. The increased amount of MDSC correlated with higher minimal residual disease in chemotherapy-treated AML patients [214].

In vivo, TKIs - imatinib and nilotinib, decreased the half-life of circulating monocytes, as well as their recruitment to tissues and differentiation into macrophages, which might reduce the generation of tumor-promoting macrophages [215]. Another study showed that imatinib prevented the polarization of macrophages toward a M2-like phenotype, induced by IL-13 (interleukin 13) or IL-4 (interleukin 4) in vitro, which was illustrated by the reduced expression of surface marker CD206 and genes such as *Arg1, Mgl2, Mrc1, CDH1,* and *CCL2* [216]. However, though in patients after TKI therapy the percentage of CD68+, CD163+, and CD206+ macrophages have decreased, they was still higher compared to the healthy control group [204]. Therapy with TKIs has also led to significant decrease of Arg1 concentration in plasma of patients. As before start of this therapy, concentration of this enzyme correlated with the number of CD11b+CD14-CD33+ myeloid cells, this decrease might be attributed to decrease in MDSC numbers [122].

Amount of regulatory T cells (Treg) has decreased in BM of AML patients who responded to chemotherapy, but failed to decrease in non-responding patients [217]. Also combined treatment with chemotherapy and histone deacetylase (HDAC) inhibitor panobinostat led to decreased number of TNFR2+ Treg in blood and BM of AML patients [218]. In CML, Treg numbers in blood decreased in patients who responded to dasatinib treatment [219], as well as imatinib and nilotinib [211]. However, suppressive phenotype (expression of CTLA-4, GITR, IL-4 and IL-10) of Treg only decreased in imatinib and dasatinib treated patients, whereas nilotinib did not induce a significant effect [211]. Importantly, amount of Treg in blood of patients has achieved same level as in healthy controls in CML patients who achieved treatment-free remission [220]. This underlines significance of Treg for successful treatment free remission and lack of leukemia relapse. Interestingly, regulatory T cells can be directly affected and targeted by imatinib [210,221,222]. This is due to an off-target activity of imatinib towards LCK (lymphocyte-specific protein tyrosine kinase), a kinase engaged in TCR signal transduction in T cells. As Treg cells generally express low levels of LCK, they are prone to apoptosis after inhibition of this kinase and thereby TCR-driven activating pathways [221]. 

Finally, first clinical studies have provided evidence that targeting the immune system may be beneficial in myeloid leukemias. Combination of azacitidine and nivolumab (anti-PD-1 antibody) was safe and provided promising overall survival and response rate in relapsed/refractory AML patients [223]. Several other clinical trials in both CML and AML have been conducted or are ongoing, to evaluate if targeting mechanisms of immune evasion may be beneficial for treatment of these cancers, either as monotherapy or in combination with other drugs (Table 2) [224].

## 7. Conclusions and Further Directions

Myeloid leukemias create an immune evasive microenvironment that promotes disease development and progression. Involvement of dynamically migrating cells, soluble factors and extracellular vesicles suggests that modulation of immune system may be a relevant for spreading of CML and AML cells outside the bone marrow - into the bloodstream and secondary leukemic niches (such as the spleen). Importantly, this immunosuppressive microenvironment seems to reverse after successful therapy, whether with tyrosine kinase inhibitors or chemotherapy, suggesting that reinvigoration of immune response constitutes a relevant factor for recovery of patients. At the same time, this underlines potential of targeting the immune system, immunosuppressive cells and factors in patients who failed to respond to first-line treatments. 

However, to accomplish this, more detailed studies are needed, to very precisely identify biomarkers of immunosuppression in myeloid leukemias and specific factors and receptors that could be targeted to e.g. eliminate suppressive immune cells. For example, most studies on Treg in CML/AML have been performed using only relatively simple phenotyping based on expression of CD25, CD127 and Foxp3. Few studies to date have employed newly proposed strategies to differentiate between naïve/effector/non-suppressive Treg [173], Treg of different origin and activity, based on markers such as Helios, GPA33 (glycoprotein A33) or CD31, or, finally, different epigenetic signatures and novel functional markers [225,226]. Similarly, MDSC and macrophages have been predominantly studied with well-recognized markers of differential polarization of myeloid cells, which is limited and may even attribute cells to “artificial”, pre-defined subsets. Such approach may be imprecise and not beneficial, taking into account plasticity of myeloid cells and existence of multiple transitional stages. New subsets of tumor-promoting myeloid cells are constantly discovered, such as recently described TREM2(triggering receptor expressed on myeloid cells 2)-expressing cells [227,228]. Finally, still very little is known about mechanisms that drive immunosuppressive cells in leukemia, especially in CML. This limits possible therapeutic targets and markers, which could be used as early predictors of leukemia development or progression. In AML, studies linking different genetic backgrounds to immune signature of the disease are still lacking, which limits development of precise targeted therapies combined with immunotherapies.

All this perhaps comes down to performing high-resolution studies of myeloid leukemias – using novel technologies, such as single cell genomics, proteomics and high-resolution spatial analyses. These have allowed for huge progress in knowledge of solid tumors and hopefully will soon allow to introduce new therapeutic modalities. Using these advanced tools on clinical specimens from leukemia patients and combining them with advanced immunocompetent mouse models of CML/AML should enable significant progress of the field.

## Figures and Tables

**Figure 1 cancers-13-01203-f001:**
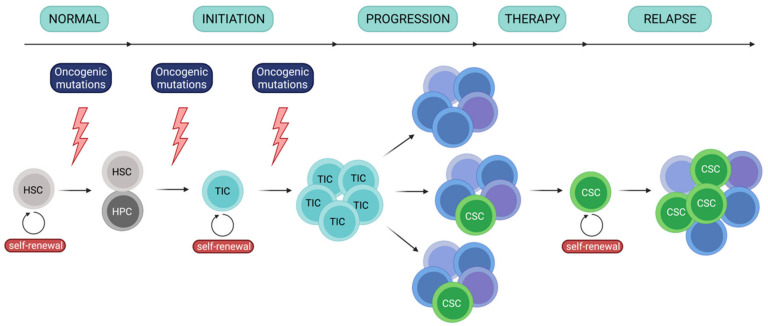
Scheme of leukemia initiation, development and relapse. Leukemia develops from tumor initiating cells, following oncogenic mutations. As leukemia develops, small population of cancer/ leukemia stem cells is maintained and enables disease relapse after therapy. HSC - hematopoietic stem cell; HPC - hematopoietic progenitor cell; TIC - tumor initiating cell; CSC - cancer stem cell.

**Figure 2 cancers-13-01203-f002:**
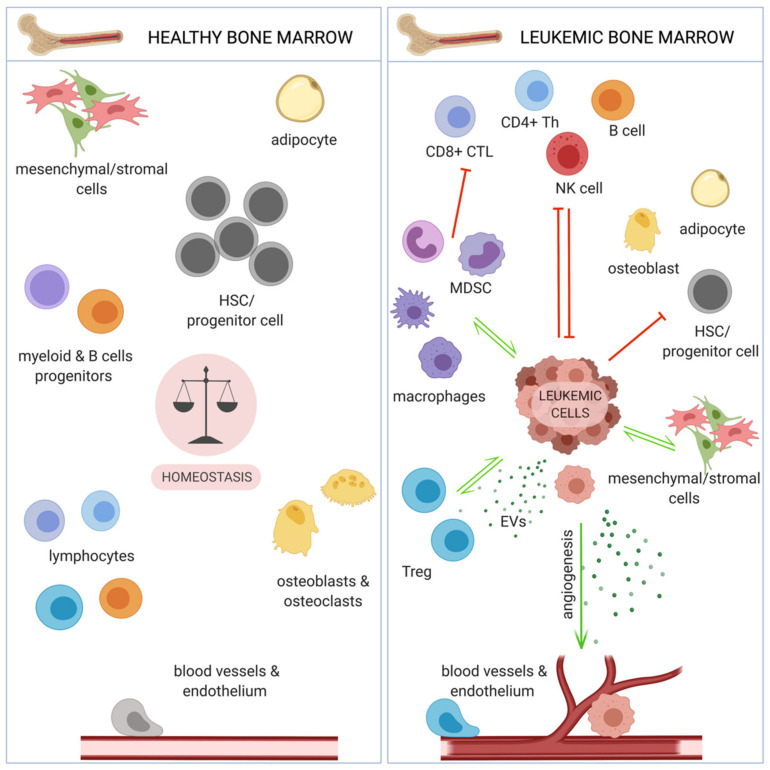
Bone marrow microenvironment in steady state and myeloid leukemias. While healthy bone marrow is characterized by well-balanced homeostasis (left panel), leukemic bone marrow is out of balance (right panel). Several cell types and processes that facilitate leukemia development are strongly promoted (green arrows), and some physiological processes, including normal hematopoiesis and immune response, are inhibited (red arrows). HSC - hematopoietic stem cell; CTL - cytotoxic T lymphocyte; NK - natural killer; MDSC - myeloid derived suppressor cell; Treg - regulatory T cell; EVs - extracellular vesicles.

**Figure 3 cancers-13-01203-f003:**
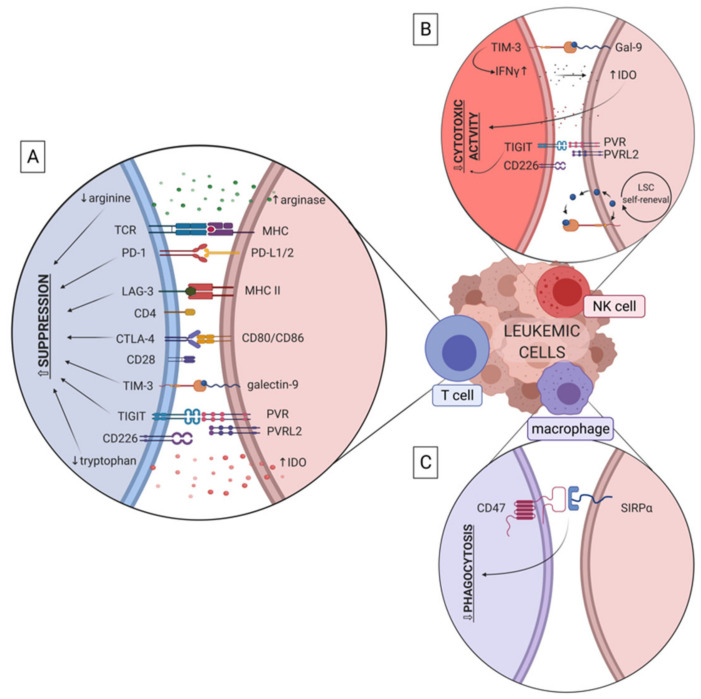
Receptors and factors expressed/released by myeloid leukemia cells to evade effector immune response (T cells, NK cells, macrophages). Leukemic cells (in brown), and their interactions with T cells (blue, panel A), NK cells (red, panel B) and macrophages (violet, panel C) are presented. TCR - T cell receptor; MHC - major histocompatibility complex; IFN-γ - interferon γ; LSC - leukemia stem cell. Other abbreviations are explained in subsections (3.1.-3.9.) of chapter 3.

**Figure 4 cancers-13-01203-f004:**
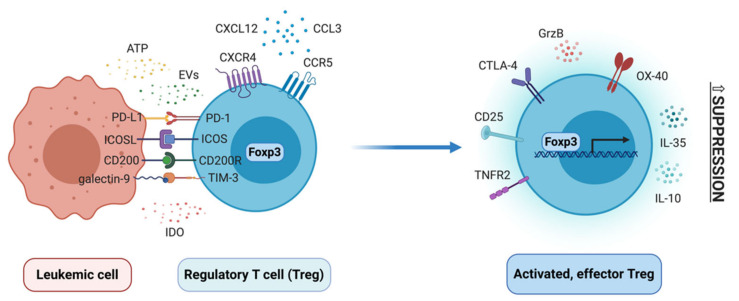
Biology of regulatory T cells in myeloid leukemias. Several receptors and factors expressed and released by leukemic cells, following interaction with their respective receptors/ligand, may drive differentiation of activated, effector Treg. Treg homing to bone marrow may be induced by chemokines, such as CXCL12 or CCL3. Activated, effector Treg in myeloid leukemias upregulate several suppressive markers and inhibit effector immune response. ATP - adenosine triphosphate; EVs - extracellular vesicles; PD-1 - programmed cell death protein 1; PD-L1 - programmed death ligand-1; ICOS - inducible T-cell costimulator; ICOSL - ICOS ligand; TIM-3 - T cell immunoglobulin and mucin domain 3; IDO - indoleamine 2,3-dioxygenase; Foxp3 - forkhead box P3; CXCL12 - C-X-C motif chemokine 12; CCL3 - chemokine (C-C motif) ligand 3; CXCR4 - C-X-C chemokine receptor type 4; CCR5 - C-C chemokine receptor type 5; TNFR2 - tumor necrosis factor receptor 2; CTLA-4 - cytotoxic T-lymphocyte-associated protein 4; GrzB - granzyme B; IL-35 - interleukin 35; IL-10 - interleukin 10.

**Figure 5 cancers-13-01203-f005:**
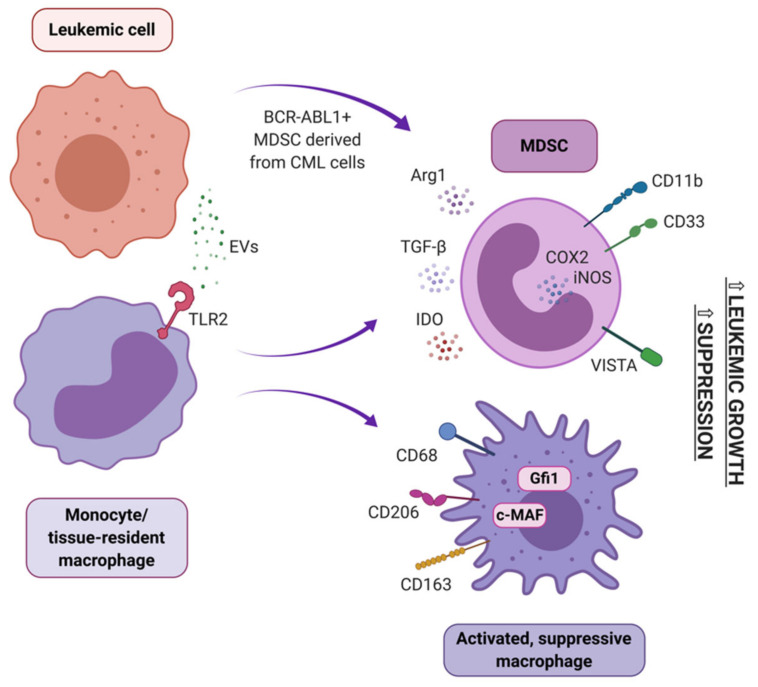
Origin and activity of highly suppressive MDSC and macrophages in myeloid leukemia – immune cells that suppress effector immune response, as well as directly support leukemic cell growth. Monocytes and tissue-resident macrophages can polarize towards either suppressive macrophages or myeloid derived suppressor cells (MDSC), e.g. due to influence of leukemic EVs. MDSC can also differentiate from BCR-ABL1+ chronic myeloid leukemia cells. Suppressive macrophages and MDSC express variety of factors (either soluble or receptors, indicated on each type of cell) that hamper effector cells function and support leukemic growth. EVs - extracellular vesicles; TLR2 - toll-like receptor 2; Arg1 - arginase-1; TGF-β - transforming growth factor β; IDO - indoleamine 2,3-dioxygenase; COX2 - cyclooxygenase 2; iNOS - inducible nitric oxide synthase; VISTA - V-domain Ig suppressor of T cell activation; Gfi1 - Growth factor independence 1.

**Table 2 cancers-13-01203-t002:** Clinical trials of drugs and treatments that target immunosuppressive factors (either as monotherapy or in combination) in chronic and acute myeloid leukemia, either ongoing or completed trials, according to clinicaltrials.gov database [224].

Target	Drug	Phase	Trial Number (clinicaltrials.gov)	Malignancy
PD-L1	Avelumab	1–2	NCT03390296	AML
		1–2	NCT02767063	CML
	Atezolizumab	1	NCT02892318, NCT03922477	AML
		1–2	NCT03730012	AML
	Durvalumab	2	NCT02775903	AML
PD-1	Nivolumab	1	NCT01822509	CML, AML
		1	NCT02011945	CML
		1	NCT04361058	AML
		1–2	NCT03825367	AML
		2	NCT02397720, NCT02464657, NCT02275533, NCT02532231	AML
	Pembrolizumab	1	NCT02981914, NCT03969446, NCT03286114	AML
		1–2	NCT03761914, NCT02996474	AML
		2	NCT03769532, NCT02768792 NCT02845297, NCT04284787, NCT02708641	AML
	PDR001	1	NCT03066648	AML
	Tislelizumab	2	NCT04541277	AML
CTLA-4	Ipilimumab	1	NCT00060372	CML, AML
		1	NCT03912064, NCT01757639 NCT01757639, NCT02890329	AML
PD-1 + CTLA-4	Nivolumab + ipilimumab	1	NCT01822509, NCT03600155	CML, AML
		2	NCT02397720	AML
TIM-3	MBG453	1	NCT03940352, NCT03066648	AML
		2	NCT04150029	AML
CD47	ALX148	1–2	NCT04755244	AML
	Magrolimab	1–2	NCT04435691	AML
	IBI188	1–2	NCT04485052	AML
	TJ011133	1–2	NCT04202003	AML
CD25 (Treg)	ADCT-301	2	NCT04639024	AML
arginine	PEG-BCT-100 (recombinant arginase 1)	2	NCT02899286	AML

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
