# Peer review of "Immunosuppressive Cell Subsets and Factors in Myeloid Leukemias"

_cancers, 2021, doi:10.3390/cancers13061203_

Round 1

Reviewer 1 Report

The story of Julian Swatleret al. is a comprehensive review on the main immunosuppressive features of myeloid leukemias. The paper is well written and informative; as such it deserves the publication in Cancers. Below, I enclose a few comments that can potentially help to further improve/strengthen its message: 

  • the distinction between tumor initiating cells and tumor stem cells is a matter of many misunderstandings. As the Authors comprehensibly cover this point, I would suggest the introduction of the diagram that would depict the differences between TICs and CSCs;
  • the concept of "clonal evolution of CSCs" should be emphasized as a consequence of CSC drug-resistance/heterogeneity;
  • About 20% of AML cases are characterized by absence of neoplastic CD34+ cells. In this case commonly small CD34+ (<1%) blast population does not contain Leukemia Stem Cells. By definition, these CD34- patients lack in CD34+CD38- and CD34+CD38+ leukemic populations. How do the authors comment this sentence?
  • LSCs in AML have not been universally identified and it is highly likely that LSCs are quite heterogeneous and the phenotypic identification could be including different markers combinations depending on the AML subtype. Can the authors provide a table that contains a list of the markers used?
  • The use of many abbreviations, making it difficult to read. I recommend including a list of all abbreviations used in the text and paying attention to write the full names of the acronyms reported in the text. 
  • The authors should add a section of microRNAs and lncRNAs and discuss the following references of these two mirs ( miR-300 and miR-126) important for entrance and exit from quiescent state of leukemia quiescent cells and the involvement of NK cells.

(Silvestri G, Trotta R, Stramucci L, Ellis JJ, Harb JG, Neviani P, WanS, Eisfeld AK, Walker CJ, Zhang B, Srutova K, Gambacorti-Passerini C, Pineda G, Jamieson CHM, Stagno F, Vigneri P, Nteliopoulos G, May PC, Reid AG, Garzon R, Roy DC, Moutuou MM, Guimond M, Hokland P, Deininger MW, Fitzgerald G, Harman C, Dazzi F, Milojkovic D, Apperley JF, Marcucci G, Qi J, Polakova KM, Zou Y, Fan X, Baer MR, Calabretta B, Perrotti D. Persistence of Drug-Resistant Leukemic Stem Cells and Impaired NK Cell Immunity in CML Patients Depend on MIR300 Antiproliferative and PP2A-Activating Functions. Blood Cancer Discov. 2020 Jul;1(1):48-67. doi: 10.1158/0008-5472.BCD-19-0039. PMID: 32974613; PMCID: PMC7510943).      

(Zhang B, Nguyen LXT, Li L, Zhao D, Kumar B, Wu H, Lin A, Pellicano F,   Hopcroft L, Su YL, Copland M, Holyoake TL, Kuo CJ, Bhatia R, Snyder DS, Ali H, Stein AS, Brewer C, Wang H, McDonald T, Swiderski P, Troadec E, Chen CC, Dorrance A, Pullarkat V, Yuan YC, Perrotti D, Carlesso N, Forman SJ, Kortylewski M, Kuo YH, Marcucci G. Bone marrow niche trafficking of miR-126 controls the self-renewal of leukemia stem cells in chronic myelogenous leukemia. Nat Med. 2018 May;24(4):450-462. doi: 10.1038/nm.4499. Epub 2018 Mar 5. PMID: 29505034; PMCID: PMC5965294).

Author Response

Response to comments by Reviewer 1

We thank the reviewer for positive reception of our manuscript and address the comments below:

Point 1: the distinction between tumor initiating cells and tumor stem cells is a matter of many misunderstandings. As the Authors comprehensibly cover this point, I would suggest the introduction of the diagram that would depict the differences between TICs and CSCs;

Response: We have addressed this issue by both describing it in the text (lines 36-49) and adding an additional figure (Figure 1), that depicts development of leukemia from TICs and role of CSCs, e.g. in therapy resistance. 

Point 2: the concept of "clonal evolution of CSCs" should be emphasized as a consequence of CSC drug-resistance/heterogeneity;

Response: We have addressed this by describing concept of clonal evolution in the context of therapy resistance and immunogenicity of leukemic cells, citing newest single-cell DNA sequencing data (lines 182-188). 

Point 3: About 20% of AML cases are characterized by absence of neoplastic CD34+ cells. In this case commonly small CD34+ (<1%) blast population does not contain Leukemia Stem Cells. By definition, these CD34- patients lack in CD34+CD38- and CD34+CD38+ leukemic populations. How do the authors comment this sentence?

Response: We have added a whole section devoted to leukemic stem cells (lines 71-103), including description of "CD34- patients" who lack CD34+ LSCs (lines 86-94). In our opinion this suggests that LSCs differentiation arrest can take place either at the progenitor (CD34+ population) or at the precursor-like (CD34- population) stage of the differentiation. We think that biggest obstacle of such situation may be hampered monitoring of therapeutic outcomes and minimal residual disease. 

Point 4: LSCs in AML have not been universally identified and it is highly likely that LSCs are quite heterogeneous and the phenotypic identification could be including different markers combinations depending on the AML subtype. Can the authors provide a table that contains a list of the markers used?

Response: We have added a whole section devoted to leukemic stem cells (lines 71-103), including description of currently proposed markers of LSCs in CML (lines 70-83) and AML (lines 98-101). There is no current consensus on differential LSC markers between AML subtypes, thus we decided not to include such table, as it could be too conclusive, especially with few data existing on the topic. Especially that the phenotype reported in specific subtypes of AML (of different genetic background) does not exclude that such phenotype might be present also in another ones. Thus, the strong statement and showing the exclusive phenotypes for each subtype might lead to misunderstandings.  Moreover, to provide thorough reporting of data on LSCs in our review, we included phenotype of LSCs used in papers cited (such as in lines 293-294, 296,  336-337, 686). 

Point 5: The use of many abbreviations, making it difficult to read. I recommend including a list of all abbreviations used in the text and paying attention to write the full names of the acronyms reported in the text. 

Response: We have added full names of all abbreviations in the text, whenever the abbreviation has first appeared, as per "Instructions for authors" of the Cancers journal (e.g. lines 151-152, 348-349 etc., all marked in the text in "track changes" mode). As "Instructions for authors" of the Cancers journal state that no list of abbreviations in used in the journal, we refrained from adding it. However, to further facilitate reading of the manuscript, we have additionally added small lists of abbreviations and full names in figure legends, for abbreviations used in the figures, to facilitate perception of figures.  

Point 6: The authors should add a section of microRNAs and lncRNAs and discuss the following references of these two mirs ( miR-300 and miR-126) important for entrance and exit from quiescent state of leukemia quiescent cells and the involvement of NK cells.

Response: We have added included a paragraph (lines 433-454) on non-coding RNAs in the section on extracellular vesicles (line 388), as these two topics are very strongly interconnected. We thank for suggestions on papers to include in this section and have included findings from both works in the mentioned paragraph (citations number 153, 155). 

Reviewer 2 Report

Authors wrote a very detailed review about immunosuppressive cell subsets and factors in myeloid leukemias by describing and characterizing suppressive immune cells, regulatory T cells, and factors that forms immunosuppressive microenvironment in myeloid leukemias.

There are minor points that needs to be addressed. Please see below for these minor comments.

Figure 1, An addition of normal bone marrow microenvironment next to myeloid leukemia bone marrow would be better to compare both of them side by side. Abbreviations need to be explained/written out with full words in the figure legends.

Too many transitional words were used back-to-back in a single paragraph, such as in between lines 92 and 99.

In section “2. Dysfunction of effector immunity in chronic and acute myeloid leukemia”, it would be better to include a table to summarize everything to help the readers to follow the content.

Some abbreviations were not spell out in full words when they introduced in the first place, such as Treg, IDO, IRF4, Foxp3, and etc. Since there are a lot of abbreviations in this manuscript, authors can introduce a table of used abbreviations at the end.

Figure 2, 3, and 4 legends need to be improved, such as color codes, abbreviations, mechanisms, and etc.

Author Response

Response to comments by Reviewer 2

We thank the reviewer for positive reception of our manuscript and address the comments below:

Point 1: Figure 1, An addition of normal bone marrow microenvironment next to myeloid leukemia bone marrow would be better to compare both of them side by side. Abbreviations need to be explained/written out with full words in the figure legends.

Response: We thank for this suggestion, we have added a second panel to Figure 1 (Figure 2 in revised version, page 4). The new panel of healthy bone marrow demonstrates balanced homeostasis in opposition to abnormal, dysregulated cell circuits in leukemic BM. We have explained abbreviations for all the figures in revised manuscript (Fig. 1 - lines 52-53, Fig. 2 - lines 131-133; Fig. 3 - lines 251-254, Fig. 4 - lines 530-537, Fig. 5 - lines 635-638). We did the same for newly included Tables. 

Point 2: Too many transitional words were used back-to-back in a single paragraph, such as in between lines 92 and 99.

Response: We thank for pointing this out. Wherever possible, we have removed excess translational words (such as requested "however", "moreover", "similarly" in lines 148-152, "also", "also" in lines 489-491 etc. - as these words have been deleted, the change should be evident after comparison with previous manuscript version). 

Point 3:  In section “2. Dysfunction of effector immunity in chronic and acute myeloid leukemia”, it would be better to include a table to summarize everything to help the readers to follow the content.

Response: We have added a new table (Table 1, page 6) that summarizes knowledge on effector immune cells (CD8+, NK cells) in myeloid leukemias, to facilitate reading of Section 2 of the manuscript. 

Point 4: Some abbreviations were not spell out in full words when they introduced in the first place, such as Treg, IDO, IRF4, Foxp3, and etc. Since there are a lot of abbreviations in this manuscript, authors can introduce a table of used abbreviations at the end.

Response: We have added full names of all abbreviations in the text, whenever the abbreviation has first appeared, as per "Instructions for authors" of the Cancers journal (e.g. line 237-Treg, line 218-IDO, line 473-IRF4, line 418-Foxp3 etc., all marked in the text in "track changes" mode). As "Instructions for authors" of the Cancers journal state that no list of abbreviations in used in the journal, we refrained from adding it.

Point 5: Figure 2, 3, and 4 legends need to be improved, such as color codes, abbreviations, mechanisms, and etc.

Response: We have extended all Figure legends (lines 127-133; 250-254; 526-537; 629-638), explaining them in more detail (as e.g. color codes in Fig. 3, as well as A, B, C sections), written abbreviations in full words, explained in more detail mechanisms and interactions on Figures. 

Reviewer 3 Report

This is a comprehensive review comparing and contrasting immune effector cells in acute and chronic myeloid leukemias. The figures are very clear and the work is extensively referenced and up-to-date.

I have the following comments / suggestions:

  1. Perhaps include a summary table to highlight the similarities and differences between immune effectors in CML and AML?
  2. Perhaps include a table of clinical trials of immune effectors in CML and AML - e.g. Magrolimab studies in AML, Arginase inhibitor (BCT-100) in AML (UK LI-1 study), Avelumab in CML (ACTIW study, France).
  3. I may have missed it, but I don't see Figures 3 and 4 referenced in the text.

Author Response

Response to comments by Reviewer 3

We thank the reviewer for positive reception of our manuscript and address the comments below:

Point 1: Perhaps include a summary table to highlight the similarities and differences between immune effectors in CML and AML?

Response: We have added a new table (Table 1, page 6) that summarizes knowledge on effector immune cells (CD8+, NK cells), separately for CML and AML. 

Point 2: Perhaps include a table of clinical trials of immune effectors in CML and AML - e.g. Magrolimab studies in AML, Arginase inhibitor (BCT-100) in AML (UK LI-1 study), Avelumab in CML (ACTIW study, France).

Response: We thank for suggesting this and giving examples of trials. We believe such table adds knowledge to translational aspect of research on immunosuppressive cells and factors in myeloid leukemias. We have created such table based on clinicaltrials.gov database and included it in our manuscript (Table 2, page 18). 

Point 3: I may have missed it, but I don't see Figures 3 and 4 referenced in the text.

Response: We thank for pointing this out. We have now referenced both Figures (Figure 4 and 5 in new version) in the text (line 481 for Fig. 4, line 609 for Fig. 5). We made sure other figures have been referenced in the text as well.